# The Role of Neutral Sphingomyelinase-2 (NSM2) in the Control of Neutral Lipid Storage in T Cells

**DOI:** 10.3390/ijms25063247

**Published:** 2024-03-13

**Authors:** Rebekka Schempp, Janna Eilts, Marie Schöl, Maria Fernanda Grijalva Yépez, Agnes Fekete, Dominik Wigger, Fabian Schumacher, Burkhard Kleuser, Marco van Ham, Lothar Jänsch, Markus Sauer, Elita Avota

**Affiliations:** 1Institute for Virology and Immunobiology, University of Wuerzburg, 97078 Wuerzburg, Germany; rebekka.schempp@uni-wuerzburg.de (R.S.); marie.schoel@uni-wuerzburg.de (M.S.); maria.grijalva-yepez@uni-wuerzburg.de (M.F.G.Y.); 2Department of Biotechnology and Biophysics, Biocenter, University of Wuerzburg, 97074 Wuerzburg, Germany; janna.eilts@uni-wuerzburg.de (J.E.); m.sauer@uni-wuerzburg.de (M.S.); 3Pharmaceutical Biology, Julius-von-Sachs-Institute, Biocenter, University of Wuerzburg, 97082 Wuerzburg, Germany; agnes.fekete@uni-wuerzburg.de; 4Department of Pharmacology and Toxicology, Institute of Pharmacy, Freie Universitaet Berlin, 14195 Berlin, Germany; dominik.wigger@fu-berlin.de (D.W.); fabian.schumacher@fu-berlin.de (F.S.); burkhard.kleuser@fu-berlin.de (B.K.); 5Cellular Proteome Research Group, Helmholtz Centre for Infection Research, 38124 Braunschweig, Germany; marco.van-ham@helmholtz-hzi.de (M.v.H.); lothar.jaensch@helmholtz-hzi.de (L.J.)

**Keywords:** neutral sphingomyelinase-2 (NSM2), lipid droplet (LD), plasma membrane (PM), diacylglycerol (DAG), cholesteryl ester (CE), triacylglycerol (TAG), fatty acid oxidation (FAO), monounsaturated fatty acid (MUFA)

## Abstract

The accumulation of lipid droplets (LDs) and ceramides (Cer) is linked to non-alcoholic fatty liver disease (NAFLD), regularly co-existing with type 2 diabetes and decreased immune function. Chronic inflammation and increased disease severity in viral infections are the hallmarks of the obesity-related immunopathology. The upregulation of neutral sphingomyelinase-2 (NSM2) has shown to be associated with the pathology of obesity in tissues. Nevertheless, the role of sphingolipids and specifically of NSM2 in the regulation of immune cell response to a fatty acid (FA) rich environment is poorly studied. Here, we identified the presence of the LD marker protein perilipin 3 (PLIN3) in the intracellular nano-environment of NSM2 using the ascorbate peroxidase APEX2-catalyzed proximity-dependent biotin labeling method. In line with this, super-resolution structured illumination microscopy (SIM) shows NSM2 and PLIN3 co-localization in LD organelles in the presence of increased extracellular concentrations of oleic acid (OA). Furthermore, the association of enzymatically active NSM2 with isolated LDs correlates with increased Cer levels in these lipid storage organelles. NSM2 enzymatic activity is not required for NSM2 association with LDs, but negatively affects the LD numbers and cellular accumulation of long-chain unsaturated triacylglycerol (TAG) species. Concurrently, NSM2 expression promotes mitochondrial respiration and fatty acid oxidation (FAO) in response to increased OA levels, thereby shifting cells to a high energetic state. Importantly, endogenous NSM2 activity is crucial for primary human CD4^+^ T cell survival and proliferation in a FA rich environment. To conclude, our study shows a novel NSM2 intracellular localization to LDs and the role of enzymatically active NSM2 in metabolic response to enhanced FA concentrations in T cells.

## 1. Introduction

Recent studies focus on the crosstalk between tightly regulated sphingolipid homeostasis and immune cell differentiation and functionality in tumor development, infection, and autoimmune diseases [1,2,3,4]. Neutral sphingomyelinase 2 (NSM2) is a part of the sphingomyelinase pathway regulating cellular levels of ceramide (Cer). NSM2 was cloned and characterized as a brain specific sphingomyelinase [5] where its expression is the highest [6]. Its usage of sphingomyelin (SM) as a substrate for the production of Cer at neutral pH was demonstrated more than a decade ago [7]. NSM2 mediates stress and inflammation induced Cer generation [8,9,10]. It is implicated in bone mineralization, growth arrest, apoptosis, and inflammation [11,12]. The level of NSM2 transcription (SMPD3) is low in human primary T cells. However, it executes a high impact on cytoskeleton activity, physical polarization, T cell receptor (TCR) sustained signaling and energy metabolism in T cells [13,14,15,16].

Obesity is a major health problem of the 21st century. Insulin resistance and type 2 diabetes are among the related comorbidities in the individuals of an obese population. Cer are a bioactive group of sphingolipids which are implicated in the pathogenesis of obesity-related metabolic diseases [17,18]. Recently published data show that the plasma membrane (PM)-associated enzymatic activity of NSM2 is part of the insulin resistance mechanism in fat-loaded steatotic hepatocytes, where NSM2 and PM-Cer suppress insulin receptor signaling [19]. The inhibition of neutral sphingomyelinases (NSM) in skeletal muscle has been shown to ameliorate the inflammatory response to the fatty acid (FA) palmitate indicating that NSM can protect skeletal muscle cells against FA-induced insulin resistance [20].

Immunopathogenesis is associated with comorbidity in obesity, i.e., the efficiency to mount the immune response is decreased. This goes along with low-grade chronic inflammation in adipose tissue, which is a central risk factor for the development of insulin resistance and type 2 diabetes in obese individuals [21]. Obesity is associated with increased numbers of CD3^+^ T cells and macrophages infiltrating adipose tissue. IFNγ secreted by CD3^+^CD4^+^ T cells sustains inflammation, whereas CD3^+^CD8^+^ T cells promote macrophage differentiation and chemotaxis to adipose tissue. Immune cells infiltrating the adipose tissue are exposed not only to pro-inflammatory cytokines (TNFα, IL-6, MCP-1) and mechanical stress, but also to increasing concentrations of free FAs. However, the direct effect of FAs in triggering the obesity-induced inflammation in T cells is poorly understood. One of the few published studies indicate a relationship between increased SMPD3 expression levels and inflammation in individuals with high liver fat content [22].

The accumulation of lipid droplets (LDs) is one of the phenotypical hallmarks for dysregulated FA homeostasis in adipose tissue. These specialized intracellular lipid storage organelles accumulate FAs in the form of neutral lipids. Cholesteryl ester (CE) and triacylglycerol (TAG) are the main classes of neutral lipids stored in LDs. Interestingly, a few studies describe NSM2-dependent regulation of LD formation in different cell types. One of them showed increased tumor growth and accumulation of LDs in liver tumor tissue of the *fro*/*fro* mice deficient for NSM2 activity [23]. Another published data showed the role of NSM2 and Cer in hyperosmolarity-induced LD formation in human corneal epithelial cells [24]. Published data indicate a potential role of NSM2 in the regulation of cellular levels of neutral lipids and LDs in different tissues.

Here, we show a direct NSM2 association with LDs in T cells. We measured NSM activity and enhanced Cer levels in LDs isolated from T cells stably expressing an enzymatically active version of NSM2. Overexpression of the enzymatically inactive mutant NSM2-H639A did not promote Cer accumulation in LDs, indicating that NSM2 plays a role in regulating Cer content in the LD membranes.

The cellular factors regulating accumulation of LDs in T cells are largely uncharacterized. We describe the negative correlation between NSM2 enzymatic activity and LD numbers in Jurkat and human primary CD4^+^ T cells. Decreased numbers of LDs correlate with a significant reduction in long-chain unsaturated TAG levels in oleic acid (OA)-loaded Jurkat cells overexpressing enzymatically active NSM2 cells. In contrast, the neutral lipid content of LDs is not affected by NSM2 activity.

LDs are important lipid storage organelles. FAs released from LDs are used for energy production by mitochondria via fatty acid β-oxidation (FAO) and the citric acid cycle [25]. T cell development and functions are regulated through different pathways of fatty acid metabolism. A well-characterized example is the regulatory T cell compartment where cellular metabolism and functionality is dependent on FAO. Specifically, Foxp3 expression is regulated through TAG storage in LDs, which promotes Foxp3 induction in response to TGFβ [26]. The contribution of LD biogenesis and FAO to the inflammatory state of T cells in adipose tissue is largely uncharacterized. We have shown previously that NSM2 facilitates the mitochondrial energy metabolism [14]. Now we show that NSM2 activity promotes fatty acid usage for mitochondrial FAO in response to increased OA levels in Jurkat cells. Our data indicate the potential positive regulatory function of NSM2 in supporting enhanced mitochondrial FAO in obese cell culture conditions. Furthermore, we substantiated the role of NSM2 in suppressing LD accumulation and supporting mitochondrial energy metabolism in primary human CD4^+^ T cells. This work defines NSM2 as a cellular factor supporting T cell survival and proliferation in a FA rich environment.

## 2. Results

### 2.1. LD Proteins Are Detected in the Nano-Environment of NSM2

Previously, we have shown that NSM2 activity at the PM regulates the generation of neutral lipids, i.e., CEs, thereby promoting sustained T cell proliferation [27]. Here, we aimed to characterize the protein environment of NSM2 to identify the cellular factors associated with neutral lipid homeostasis in T cells. Therefore, we applied the proximity-dependent biotin labeling method using the engineered ascorbate peroxidase APEX2 combined with protein mass spectrometry to analyze proteins in close proximity to NSM2 in Jurkat cells stably expressing NSM2-APEX2 (Figure 1A).

Bioinformatical analysis of the obtained proteomic data identified 3143 significantly enriched proteins (Appendix A). We selected 34 proteins which were significantly and highly enriched (by more than 3.5-fold) in NSM2-APEX2 expressing cells treated with H_2_O_2_ as compared to untreated cells (Figure 1B, Appendix A). We performed gene ontology (GO) analysis of significantly enriched biological processes within this group and identified proteins related to multivesicular body assembly and organization as well as proteins involved in exosome secretion (Figure 1C, Appendix A). Indeed, the results confirmed published data on NSM2 in the control of multivesicular body acidification and exosome secretion [28]. Interestingly, GO analysis revealed enriched proteins associated with processes of triglyceride sequestration and lipid storage: actin bundling L-plastin (LCP1) and the LD associated protein perilipin-3 (PLIN3). PLIN3 is a structural component of LDs required for their formation and maintenance. When comparing proteins identified in our proteomic analysis with the proteome of LDs isolated from human macrophages [29], we could identify another LD associated protein in the proximity of NSM2: transgelin-2 (TAGLN2) (Appendix A). Like LCP1, TAGLN2 belongs to the group of actin-binding proteins.

Taken together, NSM2-APEX2 proximity analyses identified LD associated proteins in the intracellular nano-environment of NSM2. Next, we focused on the analysis of NSM2 and PLIN3 intracellular localization and functioning to validate and mechanistically substantiate our proteome findings.

### 2.2. Intracellular NSM2 Is Not Associated with the ER or Organelles of the Secretory Pathway in Jurkat Cells

We previously described decreased Cer/SM ratios in PM fractions isolated from NSM2-deficient Jurkat cells, confirming PM associated NSM2 sphingomyelinase activity in this T cell line [27]. Therefore, it was unexpected to identify LD associated proteins in NSM2 proximal proteome analysis. First, we aimed to analyze the subcellular distribution of NSM2 to validate the findings. To achieve that, we generated stable Jurkat cell lines expressing GFP-tagged proteins of either enzymatically active wild-type human NSM2 or enzymatically dead mutant with single amino acid mutation at the position 639 replacing histidine to alanine in the sphingomyelinase catalytic center: NSM2-H639A (further on referred to as H639A). We verified the enzymatic activity of NSM2-GFP fusion protein in cell lysates. As expected, cells expressing wild-type NSM2 had significantly enhanced NSM activity as compared to control and H639A cells but did not affect the activity of acid sphingomyelinase (ASM) (Figure 2A). This observation correlated with increased Cer levels and significantly, albeit less pronounced, decreased SM levels in the total cell extracts and specifically in the isolated PM fraction (Figure 2B, upper graphs). Interestingly, NSM2 expression also affected the Cer/SM ratio in the organelle fraction of NSM2 expressing cells (Figure 2B, bottom graphs), implying the intracellular NSM2 localization and association with intracellular organelles. To confirm that, we performed Western blot analysis to analyze the GFP signal in isolated organelle and PM fractions from NSM2 and H639A expressing cells. Even so, the majority of GFP signal was detected in Src kinase Lck positive PM fractions, while a substantial amount of NSM2 was detected in organelles (Figure 2C). Next, we wanted to specify the intracellular localization of NSM2 in Jurkat cells. To enhance the sensitivity of microscopy analysis, we sorted cells for expression of high GFP levels. Cell fractionation data were substantiated via clearly detectable intracellular NSM2-GFP fluorescence (Figure 2D).

PLIN3 is recruited to the ER upon LD budding. We performed a co-localization analysis of NSM2-GFP and the ER membrane protein calnexin. However, we observed only a very marginal overlap of NSM2 and calnexin but no significant co-localization (Figure 2D, upper panels, Appendix A).

Stoffel et al. have shown that NSM2 is required for the canonical Golgi-mediated secretory pathway [30]. Next, we performed the co-localization analysis of NSM2 with the organelles along the secretory pathway. In contrast to published data in other cell types, NSM2 was not co-localizing with the Golgi compartment detected via giantin and GM130-specific antibodies (Figure 2D and Appendix A). Similar there was no GFP signal co-detected with the early or late endosomes positive for EEA1 and Rab7, accordingly. Analogous to this, we detected no co-localization of NSM2 and lysosomal protein LAMP1 (Figure 2D and Appendix A). Altogether, we observed a fraction of NSM2 localized in an undefined intracellular compartment.

### 2.3. NSM2 Resides in PLIN3 Positive LDs

PLIN3 is ubiquitously distributed within the cytoplasm and is recruited to the ER during the process of LD biogenesis. We performed subcellular fractionation and detected PLIN3 primarily within the cytoplasmic fraction and only minor amounts in organelle and PM fractions of Jurkat cells (Appendix A, left panel). Accordingly, super-resolution structured illumination microscopy (SIM) analysis showed a homogeneous distribution of PLIN3 in cell lines expressing either enzymatically active NSM2 or its inactive counterpart H639A (Figure 3A). However, a minor fraction of PLIN3 formed clusters or vesicle-like structures close to the PM (Figure 3A). Based on that, we speculated that some of the cytoplasmic PLIN3 is localized in LD-like structures close to the PM, which are barely present in standard cell culture conditions.

LDs are mainly found in adipose and liver tissue yet most of the cells including T cells can form LDs if they are loaded with FAs [31]. Excess FAs are converted to neutral lipids CE and TAG and stored in PLIN3 positive LDs. To promote LD formation and to validate the localization of PLIN3 in the close environment of NSM2, we supplemented the cell culture medium with mono-unsaturated FA, OA. NSM2 overexpression did not affect general FA uptake measured in cells after one hour of incubation with fluorescently labelled FA, BODIPY 555/568 C12 (Appendix A, left graph). Cell viability or PLIN3 protein expression levels were not affected either by NSM2 expression or overnight incubation with 300 µM OA (Appendix A, right graph, Appendix A). PLIN3 was recruited from the cytoplasmic to the organelle fraction, presumably LDs, upon OA treatment independent of NSM2 or H639A expression (Appendix A, right panel). Next, we compared the lipid accumulation in control cells and NSM2 or H639A expressing cells loaded overnight with OA and stained with the lipophilic dye Nile Red. Microscopic analysis showed a significant reduction in neutral lipid accumulation in cells expressing enzymatically active NSM2 (Appendix A). Interestingly, the Nile Red fluorescence signal largely overlapped with the GFP signal in cells expressing either enzymatically active or inactive versions of NSM2.

To exclude unspecific intracellular localization of overexpressed proteins in the vicinity of LDs, we analyzed GFP fluorescence in Lyn11-GFP expressing Jurkat cells. There, the first 11 residues of Lyn kinase membrane anchor, containing posttranslational myristylation and palmitoylation sites, are expressed in fusion with GFP. As expected, we detected Lyn11-GFP at the PM. Similar to NSM2, a fraction of Lyn11 showed intracellular localization. However, in contrast to NSM2, it was not detected within the sites of neutral lipid accumulation (Appendix A).

Next, we applied SIM for improved visualization of the LD structure. As expected, the majority of PLIN3 was recruited to the distinct LD-like structures in OA-loaded cells indicating LD formation (Figure 3A). Now, NSM2 was clearly co-detected with PLIN3 in ring structures surrounding the LDs (Figure 3A,B). We confirmed the presence of NSM2 in LDs isolated from OA-supplemented cells via ultracentrifugation of cell extracts (Figure 3E). Importantly, NSM2 activity was not mandatory for association with LDs and co-localization with PLIN3 (Figure 3C,E). Similarly, the size of the LDs was not affected by increased Cer levels in NSM2 overexpressing cells (Figure 3D). However, co-localization analysis of the fluorescence images and Western blot analysis of LDs revealed a reduction in H639A-GFP signal co-detected with PLIN3 (Figure 3C,E), indicating that sphingomyelinase activity can either stabilize NSM2 association with LD membranes or modulate the recruitment of PLIN3 during LD biogenesis.

### 2.4. NSM2 Activity Does Not Affect Neutral Lipid Accumulation in LDs

The ER contains only a small fraction of cellular sphingolipids meaning that ER-derived LD membranes contain low levels of SM. To investigate if LD associated NSM2 is active and can promote Cer accumulation, we isolated LDs from control and NSM2 or H639A expressing cells supplemented with OA to quantify NSM activity and lipid content. We measured increased NSM activity and Cer/SM ratio in LDs isolated from NSM2 expressing cells compared to LDs isolated from control or H639A cells (Figure 4A). The results affirmed that NSM2 actively increases Cer content in the LD membranes. Next, we tested if the NSM activity in LDs would affect the general lipid content within the LDs. Therefore, we performed mass spectrometry-based lipid analysis of isolated LDs and compared the relative levels of LD membrane lipids phosphatidylcholine (PC), monoacylglycerol (MAG) and of the two main groups of neutral lipids stored in LDs: CEs and TAGs. Measured levels of the major LD membrane lipid component PC was not significantly affected by NSM2, as well as when compared to H639A (Figure 4B, left graph). MAG can be used for the synthesis of TAG through the alternative MAG pathway by LD localized monoacylglycerol acetyltransferase 1 (MGAT) and can promote LD expansion [32]. Similarly to PC, MAG and CE levels were not affected by NSM2 activity in LDs isolated from OA-loaded cells (Figure 4B). Further on, we analyzed TAG species based on their chain length and degree of saturation. The main pool of TAG species found in LDs isolated from OA-loaded Jurkat cells was represented by long (13–21 carbon atoms) and very long-chain (more than 20 carbon atoms) unsaturated species with two to five double bonds (Figure 4C). The overall levels of measured lipids were most varying in control cells. Nevertheless, the increase in lipid content in LDs isolated from control cells was not significant if compared to NSM2 or H639A expressing cells. Interestingly, TAG levels were not significantly affected by NSM2 activity in any group of analyzed TAG species, indicating that NSM2 activity in LDs does not modulate TAG levels in LDs (Figure 4C).

As summarized in Figure 4D, very long-chain unsaturated TAG species showed the tendency of a slight increase in LDs isolated from control cells compared to LDs either from NSM2 or H639A. However, the regulation pattern was similar for NSM2 and H639A cells, suggesting that NSM2 enzymatic activity does not affect the TAG storage in LDs.

### 2.5. NSM2 Impairs Intracellular Accumulation of LDs and Neutral Lipids

According to the SIM data, the size of LDs or their subcellular localization were not apparently changed in NSM2 expressing cells (Figure 3B,D). However, we observed a significant reduction in Nile Red signal in NSM2 expressing cells (Appendix A), suggesting a negative role of NSM2 in LD accumulation. To study this finding in a more precise way, we used electron microscopy to compare the numbers of LDs accumulating in CTRL, NSM2 and H639A cells supplemented with OA overnight. The morphology of LDs (electron density, size, closeness to the nucleus, ER or peroxisomes) did not differ, but NSM2 expressing cells contained significantly less LDs. CTRL and H639A cells had around 20 LDs per cell, whereas NSM2 cells had on average less than 10 LDs (Figure 5A). Therefore, we concluded that NSM2 activity regulates the accumulation of LDs but not their neutral lipid content. Consequently, cells overexpressing NSM2 should have lower levels of total neutral lipids. To test this assumption, we measured neutral lipids in total cell extracts. CE and PC levels were not affected. In contrast, total TAG amount was strongly reduced in NSM2 expressing cells as compared to control cells or cells expressing enzymatically inactive NSM2 counterpart H639A (Figure 5B). We compared the negative effect of NSM2 activity on the accumulation of specific TAG species differing in lipid chain length and degree of saturation. The TAG levels were reduced in all chain length categories (Figure 5C, upper graphs). However, unsaturated TAG species were specifically affected by the expression of enzymatically active NSM2. NSM2 expressing cells contained more than 10 times the reduced amounts of unsaturated TAGs. In comparison, the cellular content of fully saturated TAGs was similar in NSM2 and H639A cells (Figure 5C, lower graphs). The fold change analysis of very long-chain TAG species in NSM2 cells compared to control or H639A cells showed the highest impact on unsaturated TAG species (Figure 5D). The data suggest that NSM2 activity impairs the cellular accumulation of unsaturated TAGs.

### 2.6. NSM2 Overexpression Negatively Regulates DAG Synthesis Pathways Crucial for LD Biogenesis

Next, we investigated if the TAG synthesis is negatively affected by NSM2 activity. Figure 6A depicts the scheme of the TAG synthesis pathway in the ER membrane which is a prerequisite for TAG accumulation and LD biogenesis. FA acyl-CoAs are esterified by a glycerol-3-phosphate acyltransferase (GPAT), resulting in the production of lysophosphatidic acid (LPA). Second fatty acyl-CoA is used by 1-acylglycerol-3-phosphate acyltransferase (AGPAT) to esterify LPA and form phosphatidic acid (PA) which is then dephosphorylated by PA phosphatase (PAP) to form diacylglycerol (DAG). Finally, diacylglycerol acyltransferase (DGAT) enzymes use another fatty acyl-CoA to generate TAG from DAG in the interspace of the ER membrane leaflets. First, we analyzed DGAT enzyme activity involved in the final step of TAG synthesis in the ER. We measured a small but significant increase in DGAT activity in the lysates prepared from cells after incubation in cell culture medium supplemented with OA independently of NSM2 expression (Figure 6B). However, the result did not exclude the possibility that LD biogenesis is affected by limited substrate availability for TAG synthesis.

Next, we aimed to analyze the synthesis of DGAT substrates. DAG is not only a central substrate in TAG synthesis but also exerts important signaling capacity. Therefore, it is kept at low levels and is quickly recycled. We analyzed the DAG-producing enzyme activity in the Golgi and the ER as both organelles are communicating and exchanging their membrane lipids. DAG in the Golgi is produced by sphingomyelin synthase 1 (SMS1) and phospholipase D1 (PLD1) pathways. As SMS1 is a major sphingomyelin synthase (SMS) expressed in Jurkat cells [33], we compared SMS1 protein expression and total SMS activity in Jurkat cells expressing either wild-type or enzymatic dead version of NSM2. The SMS enzymatic activity was significantly reduced in cells expressing active NSM2, whereas SMS1 protein expression was not affected (Figure 6C). Phospholipase D (PLD) enzymes generate PA which can be further converted to DAG. Enzymatic activity, but not the protein expression of PLD1 and PLD2, was impaired by the expression of Cer producing active NSM2 (Figure 6D). However, we were not able to measure PA levels to confirm that the inhibition of PLD would lead to less production of PA.

The DAG pool in the ER is regulated by a phospholipase C (PLC) dependent pathway. We detected a decrease in TCR stimulation dependent phosphorylation of PLCγ1 in cell lysates of NSM2 overexpressing cells (Figure 6E). Altogether, the data suggest reduced DAG levels in NSM2 expressing cells. The assumption was further supported by impaired protein kinase C (PKC) substrate phosphorylation measured in NSM2-GFP expressing TCR stimulated cells (Appendix A). However, we detected no NSM2 activity dependent changes in the DAG content of cellular membranes irrespective of cellular fraction (Figure 6F). DAG homeostasis is tightly regulated, and high turnover of PA and DAG could hinder to detect local fluctuations of those lipids during LD biogenesis. Therefore, the data described above cannot completely exclude the inhibitory role of NSM2 in production of ER localized DAG.

### 2.7. NSM2 Shifts FA Metabolism from Storage in LDs towards FAO in Mitochondria

LDs communicate with mitochondria via LD/mitochondria contacts and by doing so can control energy metabolism. Depending on the cell type, two modes of LD/mitochondria interaction are shown: one promoting LD expansion, and one supporting LD consumption and energy production [34,35]. FAs are an important alternative source of energy which is produced by the β-oxidation pathway in the mitochondria. During β-oxidation, FAs are metabolized to acetyl-CoA. Based on the previous observation that NSM2 activity supports mitochondria size and functionality in Jurkat cells [14], we hypothesized that NSM2 could promote FA usage by mitochondria for energy production resulting in reduced accumulation of neutral lipids. First, we compared the footprint of MitoTracker stained mitochondria in control Jurkat cells and cells expressing either NSM2 or H639A. The mitochondria area in NSM2 expressing cells was enlarged as compared to control or H639A expressing cells, supporting our previous published observations about the positive impact of NSM2 on mitochondria size (Figure 7A). Next, we performed a modified Seahorse mitochondrial stress test including acute injection of carnitine palmitoyltransferase 1 (CPT1) inhibitor etomoxir which prevents the formation of acyl carnitines and thereby the transport of FA from the cytosol into mitochondria. This allowed us to measure the oxygen consumption rate (OCR) in living cells und calculate the key parameters of mitochondrial respiration, as well as the levels of FA β-oxidation. We observed a decrease in mitochondrial oxidative phosphorylation in CTRL and H639A cells supplemented with OA overnight. Nearly all respiration parameters, such as basal and maximal respiration, spare respiratory capacity, and ATP production, were significantly decreased (Figure 7B and Appendix A). Importantly, higher accumulation levels of LDs in CTRL and H639A cells (Figure 5A) correlated with decreased levels of β-oxidation (Figure 7B, right graphs). In contrast, mitochondrial functionality in NSM2 expressing cells was not affected by OA treatment. Low cellular LD content in NSM2 cells correlated with higher β-oxidation levels as compared to CTRL or H639A expressing cells (Figure 7B). A similar pattern of regulation was observed for glycolytic activity measured as extracellular acidification rate (ECAR); OA treatment negatively affected ECAR in control and H639A cells whereas glycolytic activity in NSM2 expressing cells was not affected (Appendix A). Summarizing, OA treatment of control and H639A cells significantly impaired cell energetic homeostasis making them less energetic. Conversely, the overexpression of enzymatically active NSM2 protected the FA-loaded cells from negative effects on a cellular energetic state (Figure 7C).

### 2.8. NSM2 Promotes Survival and Proliferation of CD4^+^ T Cells in OA-Rich Environment

Previous studies described impaired T cell immune responses after being challenged by FAs [36,37,38]. CD4^+^ T cell infiltration in adipose tissue correlates with body mass index and obesity-related exhaustion [39]. The data shown above suggest a positive role of NSM2 to support a high energy state in T cells exposed to an obese environment. To investigate the role of endogenous NSM2 in the regulation of primary T cell responses in an FA rich environment, we isolated CD4^+^ T cells from the human blood of healthy donors by negative selection.

Highly sensitive NSM2 specific antibodies are not available to study the localization of scarcely expressed NSM2 in primary T cells. To investigate the role of NSM2 in neutral lipid storage and CD4^+^ T cell functionality in FA rich environment, we used the NSM2 specific inhibitor ES048. Compared to two other NSM2 inhibitors GW4869 [40] and DPTIP [41], ES048 is more efficient in the downregulation of NSM activity in human primary T cells (Appendix A) [14,42]. NSM activity was significantly reduced in ES048 treated CD4^+^ T cells at the concentration of 1.5 μM (Appendix A).

Obesity studies in mice models revealed impaired proliferative responses of T cells [43]. First, we stimulated CD4^+^ T cells with αCD3/αCD28 antibodies to induce cell proliferation in vitro. CD4^+^ T cells were stimulated in cell culture medium that was either supplemented with or without 50 µM OA for 3 days and stained with BODIPY 493/503 to analyze the capacity of T cells to accumulate the neutral lipids. Stimulated CD4^+^ T cell showed the accumulation of neutral lipid vesicles which was significantly enhanced via treatment with NSM inhibitor ES048 (Figure 8A,B, upper graph). Interestingly, the inhibition of NSM increased the accumulation of neutral lipids even in the absence of OA (Appendix A). Concurrently, the accumulation of PLIN3 positive vesicles was enhanced in CD4^+^ T cells treated with ES048 and OA as compared to OA treatment alone, indicating the regulatory role of NSM2 in LD accumulation (Appendix A).

It was shown previously that apoptosis-induced mitochondrial dysfunction can cause lipid droplet formation [44]. However, we did not observe the induction of apoptosis in the 3-day culture of CD4^+^ T cells supplemented with 50 μM OA or in combination with ES048 treatment (Figure 8B, bottom graph). Further, we cultivated αCD3/αCD28 stimulated CD4^+^ T cells up to 7 days in cell culture medium supplemented with increasing concentrations of OA alone or in combination with ES048. CD4^+^ T-cell viability was not affected by OA at 50 and 100 μM concentrations, even in combination with ES048 (Figure 8C and Appendix A). The data indicate that LD accumulation is not mediated by OA or ES048 induced apoptosis. However, the highest concentration of OA (300 μM) negatively affected cell viability after 7 days of cell culture (Appendix A). Importantly, NSM2 inhibition in CD4^+^ T cells exposed to the highest concentration of OA resulted in a strong increase in apoptosis (Appendix A, right panels) indicating the supporting function of NSM2 in CD4^+^ T-cell survival after exposure to high concentrations of OA.

We analyzed TCR ligation induced proliferation in CD4^+^ T cells labeled with CFSE dye, treated or left untreated with ES048 and cultivated for 5 days in medium supplemented with 100 µM OA. The combination of OA and ES048 treatments resulted in decreased T cell proliferation, reflected in an increase in CFSE fluorescence intensity (Figure 8D). We have shown previously the important role of NSM2 in the regulation of mitochondrial respiration in stimulated primary T cells [14]. We stimulated CD4^+^ T cells that were treated or untreated with OA alone or in combination with ES048 and compared the cell energy metabolism in a Seahorse analysis. As reported before, ES048 treatment negatively affected mitochondrial respiration and FA β-oxidation in stimulated CD4^+^ T cells, whereas OA treatment did not (Figure 8E and Appendix A).

The results suggest that NSM2 inhibition promotes lipid uptake and accumulation in CD4^+^ T cells. Concurrently, mitochondrial functionality is impaired and thereby FA conversion to energy and reduction in neutral lipid load is impaired. Taken together, our data reveal the role of NSM2 in keeping the balance between FA storage and turnover to support cellular energy demands and protect T cells against FA toxicity in an obese environment.

## 3. Discussion

Functional studies of NSM2 in T cells are proven to be very challenging. NSM2 coding SMPD3 mRNA transcript levels are low in human T cells as compared to the gene transcripts of other neutral sphingomyelinases (SMPD2, 4 or SMPD5). A lack of highly sensitive and specific antibodies impedes the detection and analysis of endogenous NSM2 protein. Nevertheless, genetic knock-down and pharmacological inhibition experiments demonstrated that, despite the relatively low transcription levels of its mRNA, NSM2 executes a high impact on T cell sphingolipid and cholesterol ratio at the PM [13,14,15,16].

Initially, NSM2 was shown to reside in the Golgi compartment of human neuroblastoma cells [5]. Further studies demonstrated the localization of the palmitoylated NSM2 at the cytosolic side of the PM where its enzymatic activity is regulated using anionic phospholipids and allosteric interdomain interactions [45,46]. Cell confluence, TNFα, PMA, and H_2_O_2_ have been shown to regulate NSM2 trafficking between the PM and the Golgi apparatus in various adherent cell lines [46,47,48]. We can confirm the NSM2 localization and enzymatic activity at the PM in non-adherent Jurkat cells [27].

Here, we used the proximity labeling strategy that exploits NSM2-APEX2 to biotinylate proteins in the nano-environment of NSM2 in living cells. The obtained mass spectrometry data provide a valuable resource for future studies on NSM2. Our GO functional analysis of the nano-environment of NSM2 revealed proteins related to exosome secretion. This confirmed the proximity labeling as valid to study the NSM2 protein environment and identify networks regulating cellular processes. Importantly, the data uncovered a novel group of NSM2 proximal proteins regulating triglyceride sequestering and lipid storage. Three proteins were significantly enriched within this group: LD residing PLIN3, cytoskeleton regulatory proteins LCP1 and TAGLN2. All three proteins are mainly localized in the cytoplasm. Published data suggest a positive role of cytoskeleton in LD formation and trafficking and several cytoskeletal proteins are identified as candidate LD proteins [49,50]. LCP1 is an actin bundling protein involved in actin filament network formation, membrane ruffling, and the regulation of TAG and cholesterol storage. The functional role of TAGLN2 in LDs is unknown. However, similar to LCP1, TAGLN2 belongs to the group of actin-binding proteins.

We reported previously that NSM2 affects sphingolipid levels mainly at the PM indicating predominant NSM2 activity there [27]. Generally, most of the proteins we found in the nano-environment of NSM2 were associated with intracellular organelles or were residing in the cytoplasm. The high proportion of intracellular proteins identified in our screening probably reflects the experimental constrain to isolate PM proteins in proteome analyses.

Experimental evaluations of the proximity labeling findings revealed a novel fraction of NSM2 residing in LDs. Major pools of NSM2 and PLIN3 are believed to be associated with separate membrane compartments in T cells: PM and intracellular lipid storage organelles, respectively. However, we detected a partial overlap in the subcellular localization of NSM2 and PLIN3 in organelle and PM fractions isolated from untreated cells. As expected, we observed PLIN3 relocalization to LDs after OA supplementation. The analysis of OA-loaded cells showed novel NSM2 association with PLIN3 positive LDs in Jurkat cells. It is plausible that NSM2 is transported to LDs after the initiation of their formation because we detected no NSM2 in ER membranes where LD biogenesis is located. The molecular mechanism of NSM2 transport to LDs is unclear and was beyond the scope of this manuscript. Nevertheless, the presence of the actin bundling proteins LCP1 and TAGLN2 in the proximity of NSM2 confirms the regulatory role of NSM2 in cytoskeleton dynamics, which is well documented in T cells [15,42]. Based on that, one can speculate that NSM2 could be transported to LDs via association with the cytoskeleton in preformed LD targeted protein complexes. Alternatively, LDs can form membrane contacts with most of the intracellular membranes including the PM where NSM2 could translocate to LD membranes.

We show that the enzymatic activity of NSM2 was not essential for the recruitment of NSM2 to LDs and did not affect the size and lipid content of LDs. However, LD numbers and cellular accumulation of TAG were strongly reduced in cells overexpressing enzymatically active NSM2. NSM2 expression in OA-loaded cells promoted mitochondrial β-oxidation and impaired FA storage in LDs. Our data suggest a positive role of NSM2 in protecting the cells from a massive accumulation of FA by activating mitochondrial FAO and increasing cellular energy production in stimulated T cells. In contrast, enzymatically inactive NSM2 mutant H639A was associated with LDs but exerted no impact on LD numbers, cellular TAG levels or FA β-oxidation.

TAG synthesis and LD biogenesis is initiated at ER membranes after FA uptake and is followed by LD maturation and growth. ER membranes are characterized by very low levels of anionic lipids. Nevertheless, those could be increased along the Kennedy pathway of TAG synthesis, which is crucial in LD biogenesis. Local dynamic increase in PA and unsaturated FAs during LD growth in the ER could support the activation of the NSM2 catalytic center, thereby increasing Cer concentrations at PM/ER contact sites or directly on LDs [5,51,52]. It was shown previously that Cer negatively affected PLD activity [53]. In line with this, NSM2-generated Cer impaired PLD activity in mice models of major depressive disorder [8]. Published data on *smpd3* −/− chondrocytes showed transcriptional upregulation of Golgi-associated SMS1 thereby affecting the NSM2-SMS1 cycle in the Golgi apparatus and DAG homeostasis [30]. Our in vitro enzymatic assays now confirm the negative impact of NSM2 on PLD and SMS activity and show impaired activation of PLCγ1, indicating a negative feedback loop between NSM2 activity and PA-DAG synthesis. Reduced phosphorylation of PKC substrates suggests decreased DAG levels in NSM2 overexpressing cells. Recently, it was shown that the TAG precursor lipids DAG and PA recruit PLIN3 to the ER membrane bilayer [54]. The suboptimal PLIN3 recruitment to Cer rich and DAG poor LDs would further impair the LD biogenesis and growth. However, our lipid analysis data did not show the impaired DAG levels in any of the subcellular fractions isolated from NSM2 expressing cells. Possibly, our measurements of DAG in total cellular or PM extracts do not reflect the high-speed dynamic fluctuations of DAG. We were not able to detect PA levels in our samples, suggesting a high turnover of PA to DAG. Alternatively, DAG levels could be replenished by TAG degradation during lipolysis initiated by TAG lipases in LDs [55]. We measured enhanced FAO in NSM2 expressing cells, as compared to parental Jurkat CTRL or H639A cells coinciding with low TAG levels. This observation can be interpreted as an enhanced turnover of neutral lipids into an energy producing pathway promoted by NSM2 activity.

Here, we hypothesized the following modes of NSM2 intervention with LD formation. NSM2-generated Cer could prevent DAG accumulation in ER membranes, resulting in impaired TAG synthesis and accumulation in the ER and PLIN3 recruitment to budding LDs. On the other hand, NSM2 activity promotes mitochondrial activity and FAO, thereby enhancing the turnover of cellular FAs. This would prevent the accumulation and storage of FAs in LDs in the form of neutral lipids and promote FA usage in mitochondrial energy production.

Finally, we evaluated the impact of NSM2 on FA homeostasis in primary human T cells expressing endogenous levels of NSM2. The pharmacological inhibition of NSM2 promoted lipid accumulation in PLIN3 positive LDs, suggesting the dampening role of endogenous NSM2 in neutral lipid accumulation in primary human T cells. As previously shown by us, mitochondrial respiration was impaired in NSM2-deficient cells [14]. Here, we show that functional impairment of mitochondria in NSM2 inhibitor treated cells was not affected by OA treatment. Suboptimal mitochondrial function in CD4^+^ T cells rendered them more susceptible to apoptosis and cell death in cell culture medium supplemented with high concentrations of OA. Importantly, antigenic stimulation-induced proliferation was suppressed in NSM2 activity inhibited cells after OA treatment at non-toxic dose. In summary, the results presented here show that NSM2 is important for the survival and proliferation of primary human CD4^+^ T cells in obese cell culture conditions.

Our data uncover a novel function of NSM2 in suppressing LD formation and promoting FAO in T cells suggesting its crucial role in obesity-associated immunopathogenesis. However, some limitations of the study should be mentioned. Spontaneous pharmacological inhibition of NSM2 in in vitro cell culture does not exclude some unrevealed unspecific effects on cellular metabolism induced using inhibitor treatment which should be further investigated. Further on, long term NSM2 overexpression in cell line can potentially modify the cellular transcriptome which could complement the direct effects of sphingolipids on neutral lipid storage and turnover. Additional studies are needed to fully understand the crosstalk between the sphingolipids within the lipid droplets and fatty acid metabolism in the adaptive immune compartment.

## 4. Materials and Methods

### 4.1. Jurkat Cell Culture

Jurkat cells were cultured in RPMI 1640/10% FCS. For PLCγ1 phosphorylation analysis cells were starved for 2 h in RPMI 1640/0.2% FCS followed by incubation with 5 µg/mL αCD3 (clone UCHT-1, BD Bioscience, Heidelberg, Germany), cross-linked with 5 µg/mL α-mouse IgG (Dianova, Hamburg, Germany) for 20 min on ice. Stimulation was initiated via cell transfer to 37 °C and stopped by adding ice-cold RIPA lysis buffer. Subsequently, cell lysates were used for Western blot analysis.

### 4.2. Generation of Stable Jurkat Cell Lines

pcDNA3.1-based NSM2-GFP and H639A-GFP expression vectors were provided by V. Kozjak-Pavlovic and used for the re-cloning of NSM2-GFP and its mutant version H639A into the pWPI lentiviral vector (TWINCORE, Braunschweig, Germany) under the promotor EF-1α for optimal expression in T cells. For the stable expression of C-terminal APEX2 fusion protein, NSM2-GFP was cloned in a pWPI-APEX2 lentiviral vector (provided by G. Gerold). PM anchor Lyn11-GFP expression vector was provided by G. Gerold. Lentiviral expression vectors of NSM2 fusion proteins were used to generate supernatants containing lentivirus particles through the transfection of HEK293 cells and subsequent transduction of Jurkat cells. GFP positive cells were selected using flow cytometry-based cell sorting (FACS) 10 days after transduction.

### 4.3. Oleic Acid Treatment

Oleic Acid (OA) medium for Jurkat and T cell culture was prepared as described previously [36,56]. Briefly, OA (Sigma-Aldrich, Taufkirchen, Germany) was dissolved in NaOH solution and complexed to FA-free BSA (Sigma-Aldrich, Taufkirchen, Germany) before it was added to RPMI 1640/10% FCS in the desired concentration. To enhance the LD content in Jurkat cells, they were incubated overnight in a cell culture medium containing 300 µM OA. Primary human CD4^+^ T cells were loaded with OA at indicated concentrations for 3 to 7 days. Cell viability was controlled using flow cytometry analysis using an APC Annexin V Apoptosis Detection Kit with PI (Biolegend, San Diego, CA, USA) according to the manufacturer’s protocol.

### 4.4. Thin-Layer Chromatography (TLC)-Based Enzyme Activity Assays

Neutral sphingomyelinase (NSM) activity (including NSM1, NSM2 and NSM3) in Jurkat cells was measured as described previously [57]. Jurkat cells were lysed through freeze/thawing (4×, liquid nitrogen) in 25 mM Tris-HCl (pH 7.4), 0.1 mM PMSF and Protease inhibitor tablet (Thermo Fisher Scientific, Waltham, MA, USA). In a total volume of 100 µL, 100 µg of post nuclear supernatant (PNS) was mixed with 50 mM Tris HCl (pH 7.4), 10 mM MgCl_2_, 0.2% Triton X-100, 10 mM DTT, 50 µM phosphatidylserine (Sigma-Aldrich, Taufkirchen, Germany), and 0.6 µM BODIPY-FL-C_12_ Sphingomyelin (N-(4,4-Difluoro-5,7-Dimethyl-4-Bora-3a,4a-Diaza-s-Indacene-3-Dodecanoyl) Sphingosyl Phosphocholine, Thermo Fisher Scientific, Waltham, MA, USA) and incubated at 37 °C for 1 h.

A previously published assay [58] was slightly modified and used to determine the enzymatic activity of acid sphingomyelinase (ASM) in cell lysates. Jurkat cells were lysed through freeze/thawing (4×, liquid nitrogen) in buffer containing 250 mM sodium acetat (pH 5.0), 0.1% NP-40, 1.3 mM EDTA and protease inhibitors (Thermo Fisher Scientific, Waltham, MA, USA). Lysates containing 100 µg of protein were mixed with 200 mM sodium acetat (pH 5.0), 500 mM NaCl, 0.2% NP40 and 0.6 µM BODIPY-FL-C_12_ Sphingomyelin (Thermo Fisher Scientific, Waltham, MA, USA) in a total volume of 100 µL and incubated at 37 °C for 1 h.

To analyze sphingomyelin synthase (SMS) activity, Jurkat cells were lysed through freeze/thawing (4×, liquid nitrogen) in 20 mM Tris-HCl (pH 7.4), 2 mM EDTA, 10 mM EGTA and protease inhibitors (Thermo Fisher Scientific, Waltham, MA, USA). Lysates containing 100 µg of protein were mixed with 10 mM Tris HCl (pH 7.4), 20 mM EDTA, 20 mM EGTA, 120 µM phosphatidylcholine (Sigma-Aldrich, Taufkirchen, Germany) and 20 µM NBD-C6-Ceramide (N-[6-[(7-nitro-2-1,3-benzoxadiazol-4-yl)amino]hexanoyl]-D-erythro-sphingosine, Avanti Polar Lipids, Alabaster, AL, USA) in a total volume of 100 µL, and incubated at 37 °C for 1 h.

All reactions were stopped with MeOH:CHCl_3_ (2:1, *v*:*v*) and subjected to a Bligh and Dyer lipid extraction [59]. Lipid extracts were spotted onto a TLC plate (Macherey-Nagel, Dueren, Germany) and developed in MeOH:CHCl_3_ (80:20, *v*:*v*). The plate was air-dried, and lipid bands were visualized and quantified using an Odyssey Fc Imaging System (LI-COR Biosciences, Bad Homburg, Germany).

### 4.5. Fluorescence-Based Enzyme Activity Assays

A published protocol for DGAT activity assay [60] was modified as described below. An amount of 15 µg of total membrane proteins were mixed with 100 mM Tris HCl (pH 7.4), 200 µM DOG (1-2-dioleoyl-sn-glycerol, Avanti Polar Lipids, Alabaster, AL, USA), 100 µM oleoyl-CoA (18:1 (n9) coenzyme A, Avanti Polar Lipids, Alabaster, AL, USA) and 10 mM MgCl_2_ in a total volume of 100 µL, and incubated at 37 °C for 30 min. The reaction was stopped with 0.1% SDS and quantified by adding CPM (7-diethylamino-3-(4-maleinimidophenyl)-4-methyl-cumarin, Sigma-Aldrich, Taufkirchen, Germany), measuring the fluorescent signal at Ex. 355 nm/Em. 460 nm.

Phospholipase D activity in cell lysates was analyzed according to the protocol provided through the PLD Assay Kit (Sigma-Aldrich, Taufkirchen, Germany). Jurkat cells were lysed through freeze/thawing (4×, liquid nitrogen) in 25 mM Tris-HCl (pH 7.4) and 0.1 mM PMSF in the presence of protease inhibitors (Thermo Fisher Scientific, Waltham, MA, USA). Lysates containing 20 µg of protein were used to determine PLD activity.

Total neutral sphingomyelinase (NSM) activity (including NSM1, NSM2 and NSM3) in primary CD4^+^ T cells and isolated LDs was determined as previously described [16,61]. Then, 1 × 10^6^ CD4^+^ T cells were disrupted through freeze/thawing (−80 °C) in NSM lysis buffer without detergents (20 mM Tris pH 7.4, 10 mM β-glycerophosphate, 5 mM DTT, protease inhibitors). Post nuclear supernatants or isolated LDs were incubated with 1.35 mM HMU-PC (6-hexadecanoylamino-4-methylumbelliferyl-phosphorylcholine, Biosynth Carbosynth, Bratislava, Slovakia) in NSM lysis buffer at 37 °C overnight (final volume 30 μL). Fluorescence was measured at Ex. 404 nm/Em. 460 nm and results were normalized to protein levels.

### 4.6. Cell Fractionation and LD Isolation

Plasma membrane (PM), organelle (Org) and cytosol (Cyt) fractions were isolated from 2.5 × 10^7^ Jurkat cells using Minute^TM^ Plasma Membrane Protein Isolation and Cell Fractionation Kit (Invent Biotechnologies, Plymouth, MN, USA) according to the manufacturer’s protocol.

LDs were isolated as previously described [62]. In brief, 2 × 10^8^ Jurkat cells were loaded with 300 µM OA overnight, washed with PBS followed by incubation in hypotonic buffer (10 mM HEPES-KOH pH 7.9, 10 mM KCl, 1.5 mM MgCl_2_, 0.5 mM DTT, 0.5 mM PMSF) and lysed via 15 strokes with a 26GA needle after 15 min of incubation on ice. The post nuclear supernatant was overlayed on a sucrose density gradient and ultra-centrifuged for 1 h, 28 000 rpm, 4 °C. The floating fat layer, containing the isolated LDs, was collected.

### 4.7. Western Blot Analysis

In total, 15 µg protein of whole cell lysates, 7 µg of PM, Org or Cyt fractions, 1.6 µg of isolated LDs were subjected to SDS-PAGE. Antibodies specific for GAPDH (1:4000) and PLD1 (1:1000) were purchased from Santa Cruz Biotechnology, Inc. (Heidelberg, Germany); pERK 1/2 (Thr202/Tyr204) (1:2000), AIF (1:1000), pPLCγ1 (Tyr783) (1:1000), p-(Ser) PKC substrate (1:1000), PLD2 (1:1000), Lck (1:1000) and β-actin (1:1000) from Cell Signaling Technology, Inc. (Leiden, Netherlands); PLIN3 (1:1500) and ACAA1 (1:1500) from Proteintech (Planegg-Martinsried, Germany); Calnexin from Sigma-Aldrich (Taufkirchen, Germany); GFP (1:500) from eBioscience^TM^ by Thermo Fisher Scientific (Waltham, USA); and SMS1 (1:1000) from Antibodies (Stockholm, Sweden). The bands were visualized using SuperSignal West Pico PLUS detection reagent (Thermo Fisher Scientific, Waltham, MA, USA) and quantified using Odyssey Fc Imaging System (LI-COR Biosciences, Bad Homburg, Germany).

### 4.8. Quantification of Lipid Uptake

Cells were incubated with 1 µM of the orange/red fluorescent fatty-acid BODIPY™ 558/568 C12 (4,4-Difluoro-5-(2-Thienyl)-4-Bora-3a,4a-Diaza-s-Indacene-3-Dodecanoic Acid (Thermo Fisher Scientific, Waltham, MA, USA) for 10 min or 1 h according to manufacturer’s protocol followed by flow cytometry analysis.

### 4.9. Neutral Lipid Analysis in Cells and LDs

Lipid extraction, sample preparation and lipid analysis using ACQUITY UPLC system coupled to a Synapt G2 HDMS qTOF-MS (all Waters, Eschborn, Germany) of isolated LDs and 2 × 10^6^ Jurkat cells were conducted according to a previously published protocol [63]. The acquisition and processing of chromatograms, peak detection and integration were performed using MassLynx and QuanLynx (version 4.1; all Waters, Eschborn, Germany).

### 4.10. Sphingolipid and DAG Analysis in Cells, PM and Organelle Fractions

Sphingolipid and DAG extraction was performed as previously described [64]. In brief, 1.5 mL methanol/chloroform (2:1, *v*:*v*) was added to the sample (1 × 10^6^ cells; PM or Org fractions) and was incubated overnight at 48 °C. C17:0 ceramide (C17:0 Cer) and d_31_-C16:0 sphingomyelin (d_31_-C16:0 SM) or d_5_-1,3-diheptadecanoin (d_5_-C17:0/C17:0 DAG) (all Avanti Polar Lipids, Alabaster, AL, USA) were added as internal standards. Samples were worked up in two separate preparations, with (for sphingolipids) or without (for DAGs) saponification of the lipid extract. The further procedure was analogous for both lipid classes. HPLC-MS/MS analyses were carried out under already published instrumental conditions [64,65]. The lipid extracts were chromatographically separated using a 1290 Infinity II HPLC (Agilent Technologies, Waldbronn, Germany) equipped with a Poroshell 120 EC-C8 column (3.0 × 150 mm, 2.7 µm; Agilent Technologies, Waldbronn, Germany). Analysis in MS/MS mode was performed using a 6495C triple-quadrupole mass spectrometer (Agilent Technologies, Waldbronn, Germany) in positive electrospray ionization mode (ESI+). The analyzed sphingolipid and DAG species, their mass transitions as well as their retention times are listed in Appendix A. Analyte peak areas were normalized to those of their corresponding internal standards followed by external calibration. Peak integration and quantification were determined using Mass-Hunter Quantitative Analysis Software (version 10.1, Agilent Technologies, Waldbronn, Germany).

### 4.11. APEX2 Proximity Labeling and Proteomics

APEX2 proximity labeling was carried out according to the published protocol from Hung et al. with optimization for Jurkat cells [66]. Briefly, 1 × 10^7^ Jurkat cells stably expressing NSM2-APEX2 were incubated in 20 mL RPMI 1640/10% FCS containing 1 mM biotin-phenol (Iris Biotech, Marktredwitz, Germany) at 37 °C for 30 min on a rotary wheel. For the labeling reaction, cells were centrifuged and resuspended in 2 mL RPMI 1640/10% FCS. H_2_O_2_ (30%, wt/wt, Sigma-Adrich, Taufkirchen, Germany) was added to a final concentration of 1 mM and cells were incubated for 1 min at 37 °C. A sample where H_2_O_2_ was omitted was included as a negative control. APEX2 labeling reaction was quenched via the addition of 20 mL ice-cold quenching solution (10 mM ascorbate, 5 mM Trolox, and 10 mM sodium azide in PBS) directly to the cell suspension. Cells were pelleted immediately (1 min, 4000 rpm, 4 °C), washed 3× with quenching solution and lysed in 100 µL RIPA buffer (50 mM Tris-HCl pH 7.5, 150 mM NaCl, 1% Triton X-100, 0.1% SDS, 0.5% Deoxycholate, 1 mM PMSF, protease inhibitor tablet (Thermo Fisher Scientific, Waltham, MA, USA) supplemented with quenchers (10 mM sodium azide, 10 mM sodium ascorbate and 5 mM Trolox) for 20 min on ice. To ensure maximum recovery of biotinylated proteins, excess of biotin phenol was removed by filtering the post nuclear lysates using 3K MWCO protein concentrator columns (Pierce, Thermo Fisher Scientific, Waltham, MA, USA). Biotinylated proteins were then enriched using 20 µL streptavidin magnetic beads (Pierce, Thermo Fisher Scientific, Waltham, MA, USA) per sample at 4 °C overnight. Next, beads were washed and stored for down-stream mass spectrometry analyses in 100 µL 100 mM TEAB. Proteins were digested on-bead using trypsin. For this, proteins were reduced for 1 h at 55 °C using a final concentration of 5 mM TCEP, followed by reduction for 30 min at RT using a final concentration of 10 mM MMTS. Trypsin was added in a ratio of 1 µg trypsin to 30 µg protein and proteins were digested overnight at 37 °C while shaking. Beads were spun down and supernatants transferred to LoBind Eppendorf tubes. Beads were resuspended in 50 µL 250 mM TEAB, spun down, and supernatants were combined and vacuum dried. Pellets were suspended in 50 µL 250 mM TEAB containing 5 mM TCEP and 10 mM MMTS and sonicated for 10 min in an ultrasound water bath. For peptide clean-up, an adapted SP3 protocol was applied [67]. For this, 20 µL carboxylate beads were added and peptides were bound overnight upon acetonitrile (ACN) addition to a final concentration of at least 95%. Beads were spun down and supernatants were transferred to new LoBind Eppendorf tubes, and additionally 20 µL of carboxylate beads was added to increase peptide recovery. Then, the beads were combined and washed twice with 100% ACN. Peptides were eluted by first using 20 µL 2% DMSO followed by a second step with 20 µL Millipore H_2_O. Purified peptides were vacuum dried, and pellets were suspended in 40 µL 0.1% formic acid and transferred to HPLC vials. For LC-MS/MS analyses, all samples were measured as triplicates on a Dionex UltiMate 3000 n-RSLC system (Thermo Fisher Scientific, Waltham, MA, USA) connected to an Orbitrap Fusion™ Tribrid™ mass spectrometer (Thermo Fisher Scientific, Waltham, MA, USA). A total of 200 ng peptides were loaded onto a C18 precolumn (3 μm RP18 beads, Acclaim, 0.075 × 20 mm), washed for 3 min at a flow rate of 6 μL/min, and separated on a 2 μm Pharmafluidics C18 analytical column at a flow rate of 300 nl/min via a linear 60 min gradient from 97% MS buffer A (0.1% FA) to 32% MS buffer B (0.1% FA, 80% ACN), followed by a 30 min gradient from 32% MS buffer B to 62% MS buffer B. The LC system was operated with the Thermo Scientific SII software embedded in the Xcalibur software suite (version 4.3.73.11, Thermo Fisher Scientific, Waltham, MA, USA). The effluent was electro-sprayed using a stainless-steel emitter (Thermo Fisher Scientific, Waltham, MA, USA). Using the Xcalibur software, the mass spectrometer was controlled and operated in the “top speed” mode, allowing the automatic selection of as many doubly and triply charged peptides as possible in a 3-s time window, and the subsequent fragmentation of these peptides. Peptide fragmentation was carried out using the higher energy collisional dissociation mode and peptides were measured in the ion trap (HCD/IT). MS/MS raw data files were processed via Proteome Discoverer 2.4 mediated searches against the human UniProtKB/SwissProt protein database (release 2021_11) using Sequest HT as search machine. The following search parameters were used: enzyme, trypsin; maximum missed cleavages, 2; fixed modification, carbamidomethylation (C); variable modification, oxidation (M); precursor mass tolerance, 7 ppm; fragment mass tolerance, 0.4 Da. The false discovery rate (FDR) was set to 0.01. To generate a final proteome list, only proteins that were identified and quantified in all three replicates with at least two unique peptides were considered. Protein regulations were calculated as ratios to the corresponding samples that were not treated with H_2_O_2_. Proteome data analysis was performed by using Perseus software (version v2.0.11, Max Planck Institute of Biochemistry, Planegg, Germany) [68] and GO Enrichment analysis by using the Enrichr platform [69,70,71].

### 4.12. Immunofluorescence Analysis

A total of 2 × 10^5^ cells per well were seeded on Lab-Tek^®^ II chamber slides for confocal microscopy or on 8 chambered cover glass (IBL, 220.140.082) for SIM, coated with 0.1 mg/mL poly-D-lysine (Thermo Fisher Scientific, Waltham, MA, USA). Cells were fixed with 2% PFA (Thermo Fisher Scientific, Waltham, MA, USA) and 0.2% glutaraldehyde (Sigma-Aldrich, Taufkirchen, Germany) for 15 min. Samples were permeabilized for 5 min with 0.1% Triton^™^ X-100. Primary antibodies: α-calnexin (1:200; Sigma-Aldrich, Taufkirchen, Germany), α-GM130 (1:100; BD Bioscience, Heidelberg, Germany), α-giantin (1:1000; BioLegend, San Diego, CA, USA), α-Rab7 (1:100; Cell Signaling Technology, Inc., Leiden, Netherlands), α-EEA1 (1:100; Cell Signaling Technology, Inc., Leiden, The Netherlands), α-LAMP1 (1:200; Cell Signaling Technology, Inc., Leiden, The Netherlands), and α-PLIN3 (1:200; Proteintech, Planegg-Martinsried, Germany) were applied followed by CF568 conjugated secondary antibodies (1:1000; Sigma-Aldrich, Taufkirchen, Germany), each incubated for 1 h at room temperature and washed three times with 1× PBS before mounting with fluorochrome G (Southern Biotech, Birmingham, AL, USA). For mitochondrial visualization, living cells were incubated with MitoTracker™ Deep Red FM (200 nM; Invitrogen, Waltham, MA, USA) for 30 min at 37 °C. To visualize LD in fixed samples, the neutral lipid dyes NileRed (50 nM; Sigma-Aldrich, Taufkirchen, Germany) and BODIPY 493/503 (4,4-Difluoro-1,3,5,7,8-Pentamethyl-4-Bora-3a,4a-Diaza-s-Indacen, 1 µM; Thermo Fisher Scientific, Waltham, MA, USA) were used. Samples were stained for 15 min at RT with no washing before mounting. Neutral lipid dye HCS LipidTOX™ Deep Red (1:200; Thermo Fisher Scientific, Waltham, MA, USA) was applied to living cells for 30 min at 37 °C before imaging.

Confocal Laser Scanning Microscopy imaging was performed using a LSM 510 Meta (Zeiss, Oberkochen, Germany), equipped with an inverted Axiovert 200 microscope and a 40× or 63× EC Plan-Apo oil objective (numerical aperture 1.3 or 1.4, respectively) and laser lines 488, 514, 561 and 633 nm. Image acquisition was performed with Zeiss LSM software 3.2 SP2.

Structured illumination microscopy (SIM) recordings were acquired with 9 phases using a Lattice SIM (Elyra 7, Zeiss Oberkochen, Germany) equipped with a Plan-Apochromat 63×/1.4 oil objective, a 488 nm laser (500 mW) and a 561 nm laser (500 mW). Emission light was projected through a beamsplitter (SBS Longpass 560) and detected using two cameras. Additionally, filter sets Bandpass (BP) 570–620 + Longpass (LP) 655 and BP 420–480 + BP 495–550, along with a laser blocking filter (LBF 405/488/561/642), were used. All images were processed in the Zeiss Zen black software (version 3.0 SR FP2) in standard strength SIM processing mode. A channel alignment was implemented during the processing step to correct for chromatic aberration and minor offset of the two different cameras. For this purpose, Z-stacks of 200 nm TetraSpecks^TM^ (Invitrogen, T14792, Waltham, MA, USA) were recorded, and SIM processed with the same settings as the images to be aligned. Subsequently, an alignment matrix was created using an affine transformation in the Zen black software and applied to the raw images. Brightness and contrast were adjusted linearly with the software Fiji (version 2.14.0/1.54f) [72] in all images. Mitochondrial Footprint and area per cell (in %) were quantified in Fiji by setting a region of interest (ROI) for each cell. Pearsons correlation analysis (PCC) and calculation of Mander’s coefficient M1 were carried out with the Fiji plugin JACoP [73] to analyze the signal colocalization. For PCC, a slice of a recorded 3D-stack was selected, and a background subtraction (rolling ball = 30 pixel) was conducted. ROIs were cut in the intracellular part of the cell to exclude PM associated NSM2-GFP signal. In total, 68 cells of each cell line from four independent experiments were analyzed. LD size measurements were conducted in Fiji using a line profile. The peak-to-peak distance from the generated k-plot was then determined and the mean value was calculated. In 10 (Jurkat NSM2-GFP) and 9 (Jurkat H639A-GFP) overview images one LD of each cell was measured.

### 4.13. Electron Microscopy

Sample preparation for electron microscopy was conducted according to a previously published protocol [74]. Transmission electron microscopy was performed at 120 kV acceleration voltage at a JEM-1400 Flash (JEOL, Freising, Germany) transmission electron microscope equipped with a Matataki 2 k × 2 k camera. LD number per cell was manually counted for each image.

### 4.14. Metabolic Flux Analysis

Metabolic flux analyses were performed using Seahorse XF96 (Agilent Technologies, Waldbronn, Germany). Jurkat cells were left untreated or loaded with 300 µM OA overnight before they were seeded at a density of 1.75 × 10^5^ cells/well in a 96 well Seahorse plate coated with 0.1 mg/mL poly-D-Lysin. Primary human CD4^+^ T cells were left untreated or preincubated with 1.5 µM ES048 diluted in cell culture medium (stock solution: 50 mM in DMSO) for 2 h before αCD3/αCD28 stimulation for 3 days. When indicated, 50 µM OA was added during stimulation. T cells were seeded at a density of 2 × 10^5^ cells/well and the plate was centrifuged for 15 min at 1450 rpm to facilitate adhesion. Seahorse XF RPMI assay medium (pH 7.4) was supplemented with 1 mM pyruvate, 2 mM glutamine and 10 mM glucose. Final concentrations of injected inhibitors were 1 µM Oligomycin, 1.5 µM FCCP (Carbonyl cyanide 4-(trifluoromethoxy)phenylhydrazone), 50 µM Etomoxir, 1 µM Rotenone and 1 µM Antimycin A, respectively. Flux analysis data was normalized to cell density. To determine FA β-oxidation, OCR after Etomoxir injection was subtracted from OCR after FCCP injection [75]. All other parameter calculations were performed according to Agilent’s protocol.

### 4.15. Primary CD4^+^ T Cell Isolation, Viability and Proliferation Assessment

Primary human peripheral blood mononuclear cells (PBMCs) were isolated from healthy donors through gradient centrifugation on Histopaque-1077 (Sigma-Aldrich, Taufkirchen, Germany). CD4^+^ T cells were enriched (≥90%) using MagniSort^TM^ Human CD4 T cell Enrichment Kit (Invitrogen by Thermo Fisher Scientific, Waltham, MA, USA) and maintained in RPMI 1640/10% FCS. For stimulation, cells were incubated with 1 µg/mL αCD3 (clone UCHT-1, BD Bioscience, Heidelberg, Germany) and αCD28 (clone CD28.2, BD Bioscience, Heidelberg, Germany) for 20 min on ice, subsequently transferred to 96-well plates precoated with 25 µg/mL α-mouse IgG (Dianova, Hamburg, Germany) (2 h at 37 °C) and incubated at 37 °C for up to 7 days. If indicated, NSM2 was inhibited through incubating the cells with 1.5 µM ES048 diluted in cell culture medium (stock solution: 50 mM in DMSO) [42] for 2 h prior to stimulation and maintained in this concentration during stimulation. To analyze proliferation, cells were labeled for 10 min with 5 µM CFSE (Affymetrix, Santa Clara, CA, USA) prior to any treatment and analyzed using flow cytometry after 5 and 7 days. The toxicity of OA and ES048 treatments was tested using an AnnexinV Apoptosis Detection Kit (Biolegend, San Diego, CA, USA) and analyzed using flow cytometry analysis.

### 4.16. Statistical Analysis

Data were acquired in at least three independent experiments involving individual donors when primary human T cells were used. Data were analyzed using GraphPad Prism software (version 10.2.1, GraphPad Software, Inc., Boston, MA, USA). For comparing two groups, a two-tailed student’s *t*-test was performed. For comparing more than two groups, one-way ANOVA with a post hoc Turkey test or two-way ANOVA with post hoc Sídák test were used. *p*-values are shown as * *p* < 0.05, ** *p* < 0.01, *** *p* < 0.001, and **** *p* < 0.0001; ns, non-significant. Bars show SDs.

## Figures and Tables

**Figure 1 ijms-25-03247-f001:**
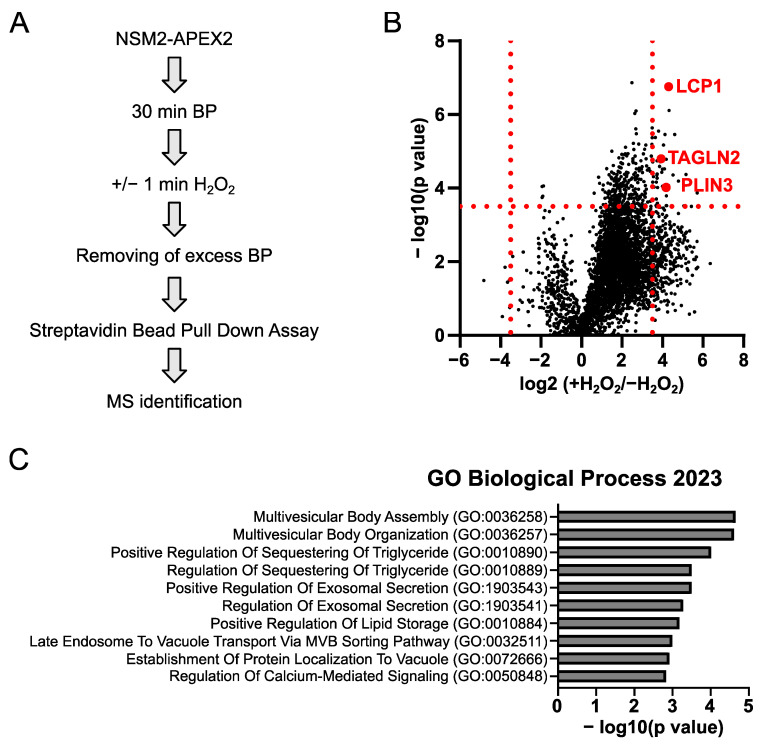
NSM2-APEX2 proximity labeling identifies proteins related to exosome secretion and neutral lipid storage. (**A**) Workflow of proximity labeling strategy in living Jurkat cells expressing NSM2-APEX2. (**B**) Volcano plot for quantitative identification of the proteins proximal to NSM2. The *x*-axis indicates the logarithm-transformed ratio of the relative abundance of a protein in NSM2-APEX2 cells incubated with H_2_O_2_ versus cells where H_2_O_2_ was omitted (*n* = 3). The *y*-axis: the logarithm-transformed *p*-value of the Student’s test. Cut offs for differentially enriched proteins (relative abundance is changed more as 3.5-fold, negative log of *p*-value more as 3.6) are marked as red dotted lines. Each black dot represents one gene. LD associated protein PLIN3, LCP1 and TAGLN2 are highlighted in red. (**C**) Gene Ontology (GO) Biological Process analysis is shown for the 34 significantly enriched proteins within upper right quadrant in B. BP: biotin phenol; H_2_O_2_: hydrogen peroxide.

**Figure 2 ijms-25-03247-f002:**
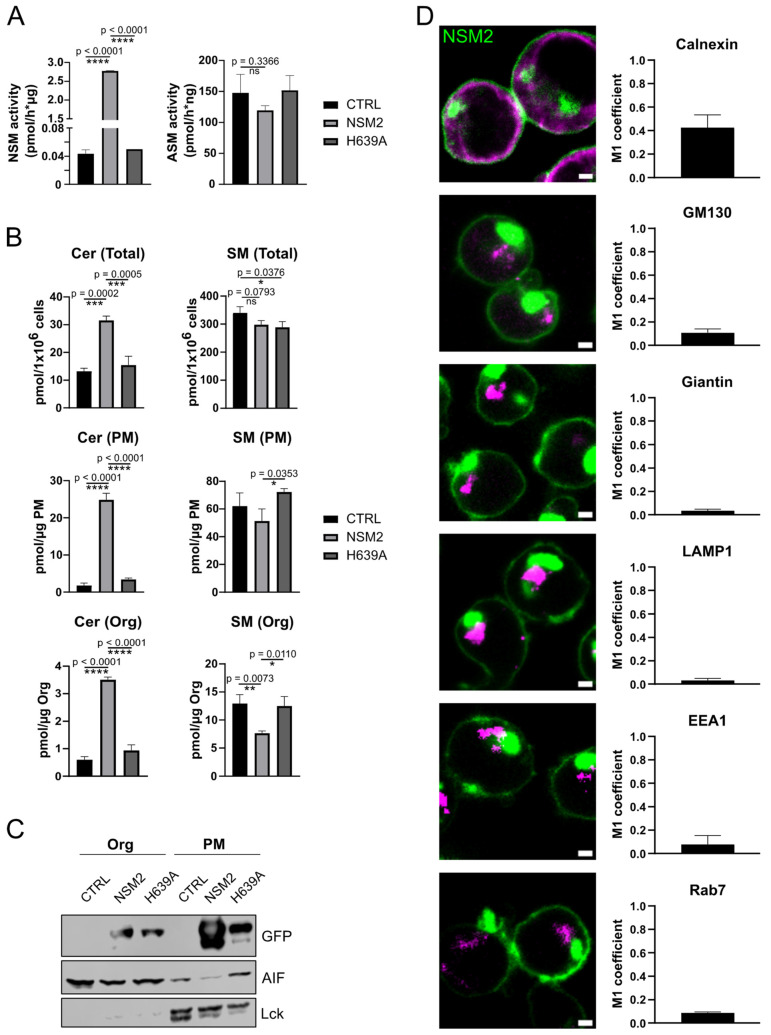
Intracellular NSM2 does not localize in the ER or organelles of the secretory pathway. (**A**) NSM (left graph) and ASM (right graph) activity in CTRL, NSM2 and H639A Jurkat cell lysates. (**B**) Cer (left column) and SM (right column) levels in lipid extracts of whole cells (upper graphs), PM (graphs in the middle) and organelle (Org) fractions (bottom graphs) of CTRL, NSM2 and H639A Jurkat cells as assessed using LC-MS/MS. (**C**) Western blot analysis of Org and PM fractions isolated from Jurkat cells. (**D**) Representative cropped fluorescence images of NSM2-GFP (green) Jurkat cells stained for different compartment markers (magenta) (left column, scale bar: 2 µm). Mander’s colocalization coefficients M1 for NSM2-GFP and each of the compartment markers are shown in the right column. Mean values with standard deviations of the measurements are shown (n = 3). *p*-values of one-way ANOVA with post hoc Turkey test are shown as * *p* < 0.05, ** *p* < 0.01, *** *p* < 0.001, and **** *p* < 0.0001; ns, non-significant.

**Figure 3 ijms-25-03247-f003:**
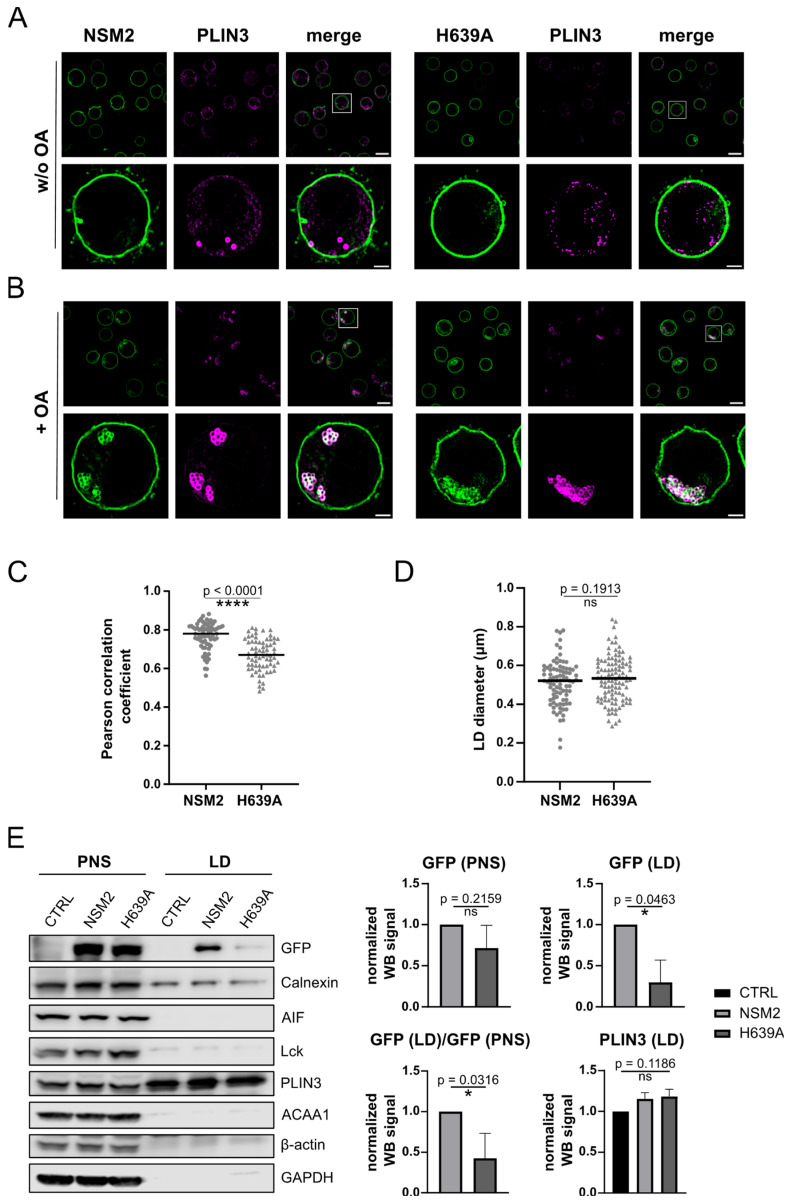
Enzymatic activity is not required for NSM2 association with lipid droplets (LDs) in oleic acid (OA)-loaded cells. Representative SIM images of NSM2- and H639A-GFP Jurkat cells left untreated (**A**) or loaded with 300 µM OA overnight (**B**) and stained for PLIN3. Scale bar: 10 µm. Rectangles indicate the location of single cell zoom in shown below (scale bar: 2 µm). (**C**) Pearson correlation analysis for NSM2 or H639A co-localization with PLIN3 in cells incubated with OA (n = 68). (**D**) LD diameter in NSM2 and H639A cells shown in B (n = 87). (**E**) Representative Western blot (**left**) and densiometric quantification (**right**) of NSM2 and different cellular compartment markers in post nuclear supernatant (PNS) and LDs isolated from OA-treated cells (n = 3). Mean values with standard deviations of the measurements are shown. *p*-values of two-tailed student’s *t*-test analysis are shown as * *p* < 0.05 and **** *p* < 0.0001; ns, non-significant.

**Figure 4 ijms-25-03247-f004:**
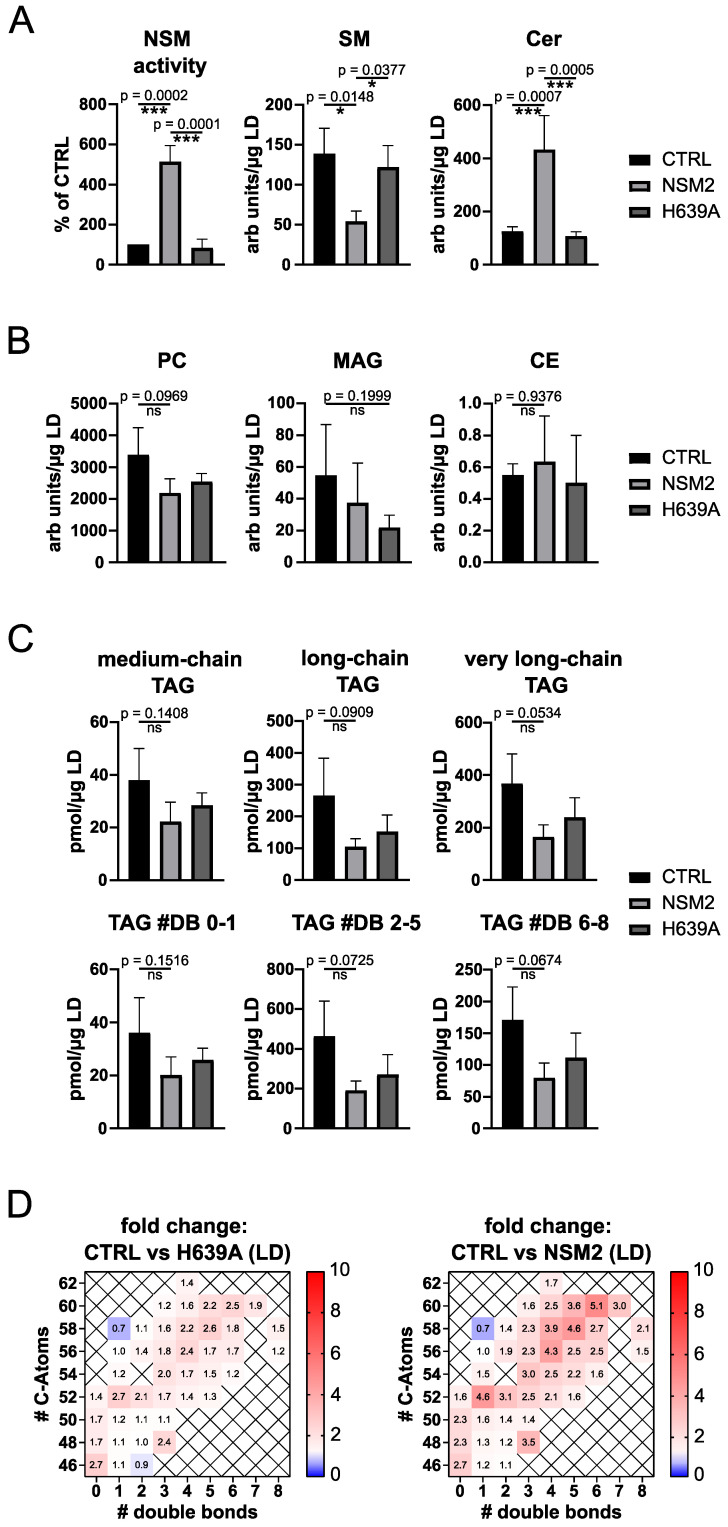
NSM2 activity in lipid droplets (LDs) does not affect their neutral lipid content. NSM activity and LC-MS lipid analysis of LDs isolated from CTRL, NSM2 and H639A cells treated with 300 µM OA overnight. (**A**) Quantification of NSM activity and sphingolipid (SM, Cer) content. (**B**) Levels of LD membrane lipids: MAG, PC, and neutral lipid: CE. (**C**) Levels of TAG species analyzed according to their acyl chain length (upper panels) or saturation (# of double bonds (DB); lower panels). Mean values with standard deviations of the measurements are shown (n = 3). *p*-values of one-way ANOVA with post hoc Turkey test are shown as * *p* < 0.05 and *** *p* < 0.001; ns, non-significant. (**D**) Heat maps of fold changes for different TAG species in LDs of CTRL vs. H639A (**left**) and LDs of CTRL vs. NSM2 (**right**). TAGs were plotted according to the number of carbons (rows) and double bonds (columns) of the FAs esterified to glycerol.

**Figure 5 ijms-25-03247-f005:**
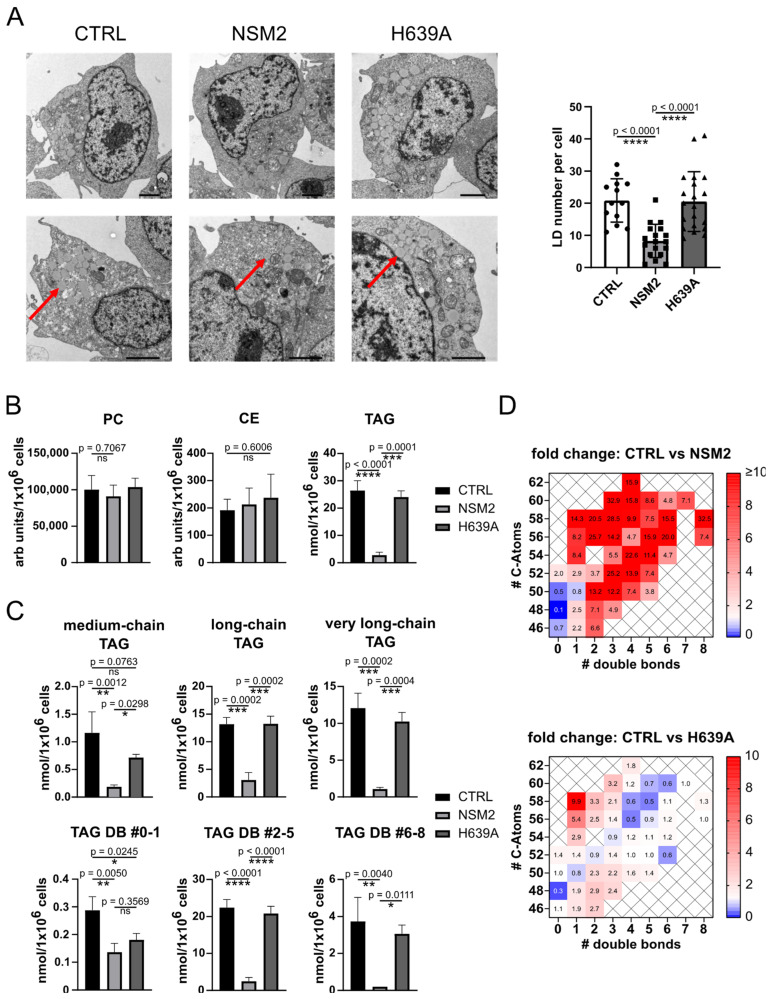
NSM2 activity impairs the cellular content of lipid droplets (LDs) and neutral lipids. (**A**) Analysis of CTRL, NSM2 and H639A cells treated with 300 µM OA overnight. Representative EM images (scale bar: 2 µm). Red arrows indicate LDs. Right graph: quantification of LD number per cell (n = 13). (**B**) Lipid analysis of TAG, PC and CE using LC-MS in total cell extracts (n = 3). (**C**) Quantification of TAG species according to their acyl chain length (**upper panels**) or saturation (# of double bonds (DB); **lower panels**). (**D**) Heat maps of fold changes for TAG species in CTRL vs. H639A Jurkat cells (**top**) and CTRL vs. NSM2 Jurkat cells (**bottom**). TAGs were plotted according to the number of carbons (rows) and double bonds (columns) of the FAs esterified to glycerol. Mean values with standard deviations of the measurements are shown. *p*-values of one-way ANOVA with post hoc Turkey test are shown as * *p* < 0.05, ** *p* < 0.01, *** *p* < 0.001, and **** *p* < 0.0001; ns, non-significant.

**Figure 6 ijms-25-03247-f006:**
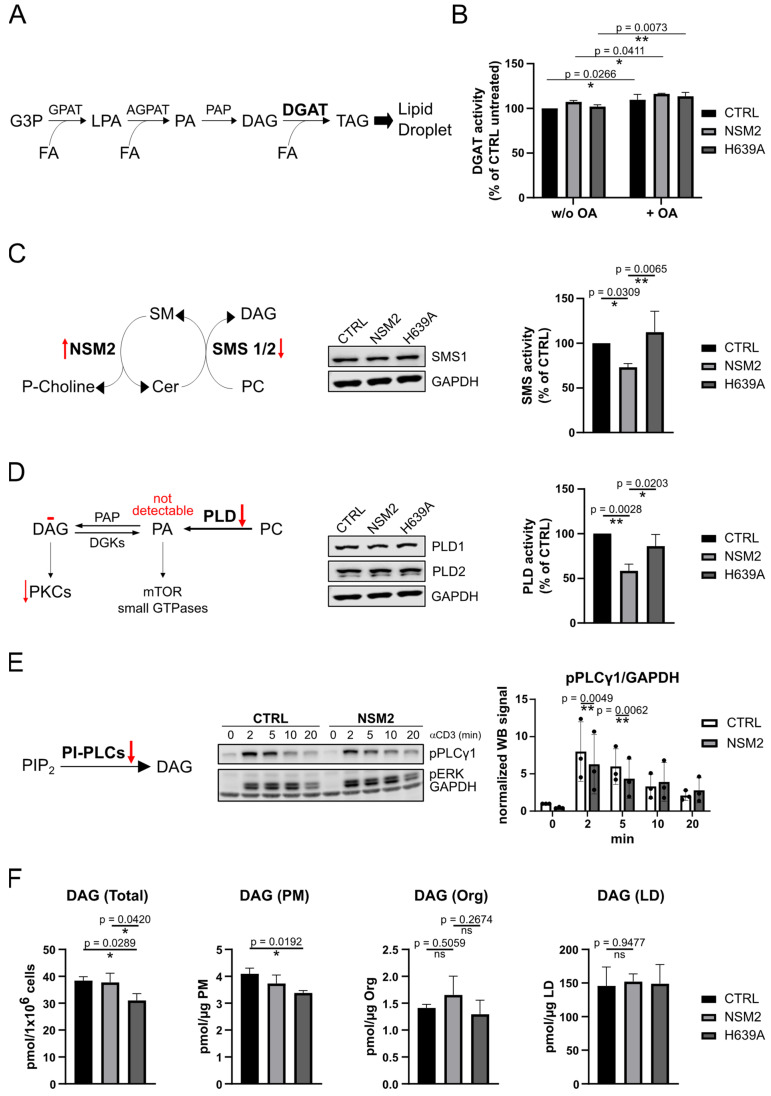
NSM2 activity negatively affects diycylglycerol (DAG) synthesis pathways upstream of neutral lipid triacylglycerol (TAG) accumulation. (**A**) Simplified scheme of the TAG synthesis pathway during LD biogenesis at the ER membrane. (**B**) Diacylglycerol acyltransferase (DGAT) activity of Jurkat cells loaded with 300 µM OA or left untreated was measured. (**C**) Sphingomyelin synthase (SMS) and (**D**) phospholipase D (PLD) enzymatic activities **(right graphs**) and protein levels (Western blot panels) in cell lysates. (**E**) Phosphorylation of phospholipase C-gamma 1 (PLCγ1) in cells stimulated with αCD3 for indicated time points. Red arrows indicate NSM2-dependent changes in enzymatic activity. (**F**) LC-MS measurements of diacylglycerol (DAG) levels in total cell extracts or subcellular fractions (PM, organelles (Org), LDs) of CTRL, NSM2 and H639A Jurkat cells. Mean values with standard deviations of three independent measurements are shown. *p*-values of one-way ANOVA with post hoc Turkey test are shown as * *p* < 0.05 and ** *p* < 0.01; ns, non-significant. DGK: DAG kinase; PIP_2_: phosphatidylinositol 4,5-bisphosphate; PI: phosphatidylinositol.

**Figure 7 ijms-25-03247-f007:**
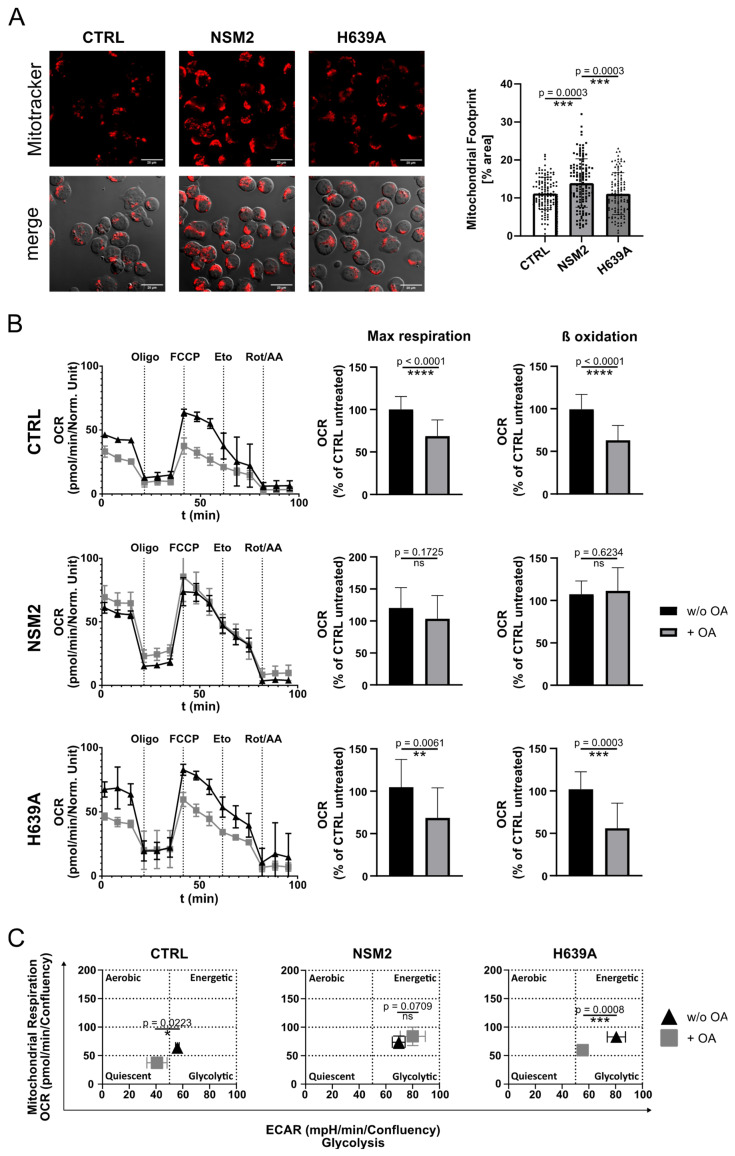
Increase in mitochondria size and fatty acid β-oxidation in cells expressing NSM2. Analysis of CTRL, NSM2 and H639A cells. (**A**) Representative fluorescence images (left panels) and quantification (right graph) of the mitochondrial footprints in individual cells (n = 107) labeled with MitoTracker™ Deep Red FM (scale bar: 20 µm). (**B**) Representative graphs of OCR in cells left untreated or treated overnight with OA (**left**). Quantification of maximal respiration and β-oxidation from three independent experiments (**right** graphs). Oligomycin (Oligo), FCCP, Etomoxir (Eto), Rotenone (Rot) and Antimycin A (AA) were injected during the measurements as indicated via the dotted lines. (**C**) Cellular energy phenotype profiles for OA-treated and untreated cells displayed as a scatter plot of OCR (*y*-axis) and ECAR (*x*-axis). Mean values with standard deviations of the measurements are shown in all graphs. *p*-values of one-way ANOVA with post hoc Turkey test (**A**) or two-tailed student’s *t*-test (**B**,**C**) are shown as * *p* < 0.05, ** *p* < 0.01, *** *p* < 0.001, and **** *p* < 0.0001; ns, non-significant.

**Figure 8 ijms-25-03247-f008:**
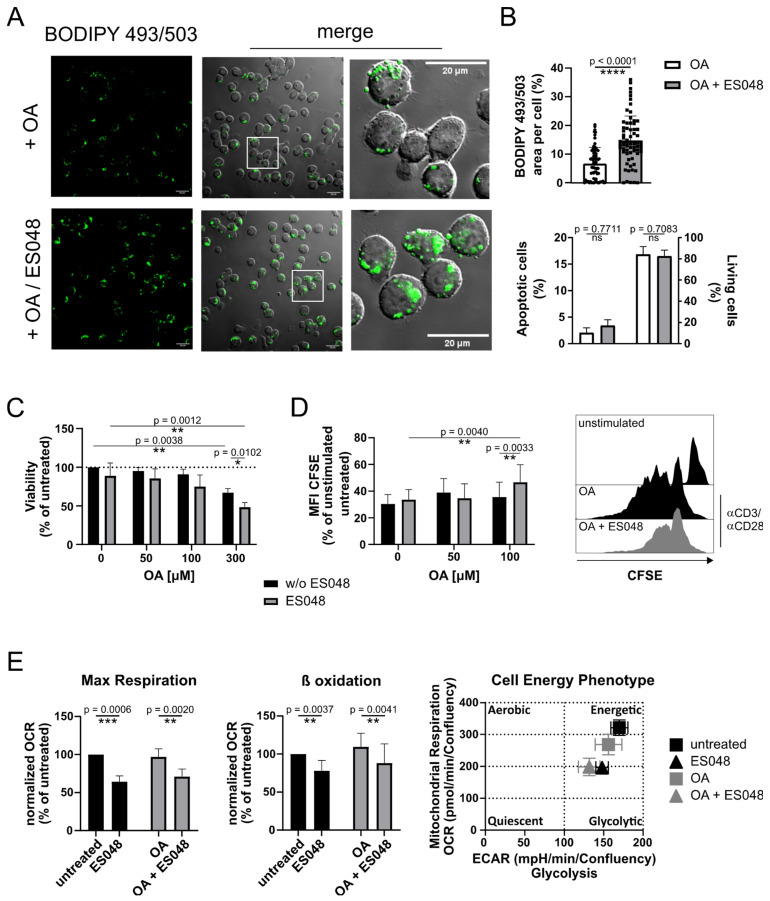
Pharmacological inhibition of endogenous NSM2 enhances lipid droplet (LD) accumulation and cell death in primary CD4^+^ T cells treated with oleic acid (OA). Human CD4^+^ T cells were left untreated or pretreated with 1.5 µM ES048 for 2 h prior to co-stimulation with αCD3/αCD28 and loading with OA. (**A**) Representative fluorescence images of cells stained with BODIPY 493/503 after 3 days of treatment with 50 μM OA. Rectangles indicate the location of zoomed areas (right panels; scale bar: 20 µm). (**B**) Flow cytometry analysis of apoptotic (AnnexinV positive) and living (Annexin V and PI negative) cells (upper graph) and quantification of BODIPY 493/503 area per cell (n = 68) (lower graph) of CD4^+^ T cells shown in A. (**C**) Viability and (**D**) proliferation of CFSE labeled CD4^+^ T cells incubated with indicated concentrations of OA for 5 days and measured using flow cytometry. Right panel shows representative CFSE profiles of the cells loaded with 100 µM OA. (**E**) Analysis of maximal mitochondrial respiration and FA β-oxidation (left graphs, n = 3). Cellular energy phenotype profiles are displayed as a scatter plot of OCR (*y*-axis) and ECAR (*x*-axis) (right panel). Oligomycin (Oligo), FCCP, Etomoxir (Eto), Rotenone (Rot) and Antimycin A (AA) were injected during the measurements (n = 3). Mean values with standard deviations of the measurements are shown. *p*-values of two-way ANOVA with post hoc Sídák test are shown as * *p* < 0.05, ** *p* < 0.01, *** *p* < 0.001, and **** *p* < 0.0001; ns, non-significant.

## Data Availability

The proteomics raw data have been deposited to the Proteomics Identifications Database (https://www.ebi.ac.uk/pride, accessed on 24 January 2024).

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
