# Peer review of "The Role of Neutral Sphingomyelinase-2 (NSM2) in the Control of Neutral Lipid Storage in T Cells"

_ijms, 2024, doi:10.3390/ijms25063247_

Round 1
Reviewer 1 Report
Comments and Suggestions for Authors
In this paper entitled “The role of Neutral sphingomyelinase-2 (NSM2) in the control of neutral lipid storage in T cells”, Schempp and colleagues explored the role of Neutral sphingomyelinase-2 (NSM2) in regulation of immune cell response to fatty acid (FA) rich environment, and the authors reported that endogenous NSM2 activity is crucial for primary human CD4+ T cell survival and proliferation in a FA rich environment. The manuscript is interesting, but there are some concerns that should be addressed before its publication.
-The acronyms used for the first time in the introduction section should be defined; for example, NSM2, Cer, and LDs. In addition, the FAO definition (fatty acid metabolism (FAO)) should be corrected.
-There are some typographical errors in the manuscript; for example, (Thermo Fisher Scientific)), 1μg trypsin, NSM.
-Why were Jurkat and CD4 T cells cultured in RPMI 1640 with 10% FCS for the oleic acid treatments and not just in RPMI 1640 without FCS? In addition, the authors should clarify how ES048 was dissolved for cell treatments.
-Please add the dilutions of the antibodies used for the experiments.
-In the results section, the western blots should be displayed with histograms, respectively (Figure S5D and Figure 8C). Please define all acronyms used in the figures (e.g., SM, etc).
-According to the findings showed in this manuscript: “We stimulated cells with αCD3/αCD28 in cell culture medium supplemented with OA and ES048 which resulted in more than 70% reduction of NSM activity (Figure 8B, upper graph). Concurrently, ES048 treated cells showed significantly enhanced accumulation of neutral lipids as compared to untreated cells (Figure 8A and B, lower panels).” In addition, the authors reported the following results: “OA at 300 μM concentration significantly reduced cell viability. However, cell death was significantly higher in ES048 treated cells (Figure 8C). Importantly, numbers of AnnexinV positive apoptotic cells were increased in cells treated with OA and ES048. More than a half of the OA-loaded and ES048 treated stimulated CD4+ T cells were apoptotic after 7 days in cell culture which led to significant reduction of viable cell counts (Figure S5D).”
The main concern of this study is that CD4+ T cells treated with ES048 show around 50% of apoptosis (Figure 8C), including a high percentage of apoptosis for OA 300 μM. With respect to these concerns; it was reported that a characteristic of apoptosis is the rapid accumulation of cytoplasmic lipid droplets, which are composed largely of neutral lipids (Apoptosis-induced mitochondrial dysfunction causes cytoplasmic lipid droplet formation, PMID: 22460322). On the other hand, the authors comments that ES048 treatment negatively affected mitochondria respiration and FA β-oxidation in CD4+ T cells, whereas OA treatment did not (Figure 8E, Figure S6). With respect to this finding, it was reported that apoptosis decreases mitochondrial fatty acid β-oxidation (Apoptosis-induced mitochondrial dysfunction causes cytoplasmic lipid droplet formation, PMID: 22460322). Therefore, the authors should repeat these experiments with low doses of ES048 or use a different NSM2 inhibitor.
-The authors should add the limitations of this study in the manuscript.
Comments on the Quality of English Language
The English language is fine. However, there are some typographical errors in the manuscript.
Reviewer 2 Report
Comments and Suggestions for Authors
The manuscript of Rebekka Schempp et al., shows a novel Neutral sphingomyelinase-2 (NSM2) intracellular localization to LDs and delineate the role of enzymatically active NSM2 in metabolic response to enhanced Fatty Acid concentrations in T cells. Authors suggested that NSM2 is a cellular factor supporting T cell survival and proliferation in a Fatty Acid rich environment. As immunopathogenesis is associated with comorbidity in obesity the topic of this manuscript is very interesting.
Methodology is clearly presented. Results are well organized and conclusions are strongly supported by the presented figures and statistics.
A minor suggestion is to summarize the findings for NSM2 intracellular involvement in a final graph. This will be of added value for the manuscript that merits to be accepted.
Author Response
I would like to thank for the positive feedback to our manuscript. Now we added the graphical abstract. All changes done in the text of manuscript are underlined and marked red.
Round 2
Reviewer 1 Report
Comments and Suggestions for Authors
I have no comments.